# LEARNING FAIR REPRESENTATIONS WITH HIGH-CONFIDENCE GUARANTEES

## ABSTRACT

Representation learning is increasingly employed to generate representations that are predictive across multiple downstream tasks. The development of representation learning algorithms that provide strong fairness guarantees is thus important because it can prevent unfairness towards disadvantaged groups for *all* downstream prediction tasks. In this paper, we formally define the problem of learning representations that are fair with high confidence. We then introduce the ***Fair Representation** learning with high-confidence **Guarantees*** (FRG) framework, which provides high-confidence guarantees for limiting unfairness across all downstream models and tasks, with user-defined upper bounds. After proving that FRG ensures fairness for all downstream models and tasks with high probability, we present empirical evaluations that demonstrate FRG's effectiveness at upper bounding unfairness for multiple downstream models and tasks.

## 1 INTRODUCTION

In every prediction task, machine learning (ML) algorithms assume two distinct roles: the data producer and the data consumer (Zemel et al., 2013; Dwork et al., 2012; Madras et al., 2018). The data consumer's role is to make accurate predictions using the data provided by the data producer. While the data producer may distribute raw data, it is also common for it to generate new representations for the input data (referred to as *representation learning*) that are predictive.

When multiple data consumers' prediction tasks involve inputs of the same type, such as natural language text or images, the data producer can generate *general representations* that are predictive to multiple subsequent tasks. With the increasing prominence of deep learning methods for tasks involving language, audio, and visual data, the adoption of these general representations has become prevalent in both academic and industrial settings. Examples of this trend include the Variational Autoencoder (VAE) (Kingma & Welling, 2013) and recent language models such as BERT (Devlin et al., 2019) and GPT-4 (OpenAI, 2023).

While representation learning can benefit various downstream predictions, it is also susceptible to the risk of producing unintended or undesirable behaviors in the downstream tasks. More precisely, the representations can be used to generate predictions that display unfairness or bias against certain disadvantaged groups. This is especially problematic in critical domains, such as loan underwriting (Byrnes, 2016), hiring (Miller, 2015) and criminal sentencing (Angwin et al., 2016), where the consequences of algorithmic bias may severely impact individuals. In these cases, fairness may mean that a person's sensitive attributes, such as race and gender, should not be pertinent to the model's predictions. Given these risks, researchers propose that fairness should be a concern not only for the data consumer that uses the representations, but also for the data producer that generates them (Zemel et al., 2013; Madras et al., 2018). A data producer that ensures fairness in the data representation ensures fairness in all downstream tasks. As a result, the data producer can release the data representation to any data consumer without concern.

To ensure the fairness of representations, simply removing sensitive attributes from the dataset is insufficient as sensitive information can still inadvertently leak through non-sensitive attributes Castelnovo et al. (2022). Numerous studies have proposed methods for learning fair representations, collectively referred to as *fair representation learning* (Zemel et al., 2013). These methods are designed to reduce the presence of sensitive information in the representations, ensuring fairness across all downstream models and tasks. In Section 8 we present a detailed discussion of related work such

as the works of Louizos et al. (2016); Madras et al. (2018); Moyer et al. (2018); Song et al. (2019); Gupta et al. (2021); Balunović et al. (2022); Kim et al. (2022) and Jovanović et al. (2023).

While much of the previous work has demonstrated effectiveness in promoting fairness in specific downstream tasks, the majority of these approaches provide little to no assurance that the unfairness of all downstream models will be consistently controlled or bounded by a user-defined error threshold with high probability. In many areas of supervised learning, providing high-confidence guarantees is considered essential for ensuring fairness, privacy, and safety of the learning algorithm (Li et al., 2022; Abadi et al., 2016; Thomas et al., 2019). This need for high-confidence guarantees becomes even more critical in the context of learning fair representation as the absence of such guarantees can lead to undesired behaviors across multiple downstream applications.

Our main contributions in this paper are as follows. We provide formal definition of learning representations that are fair with high confidence in Section 3. This definition ensures that unfairness, measured using demographic parity, is consistently upper-bounded by a user-defined threshold across all downstream predictions with high probability. Under this definition, we introduce the *Fair Representation learning with high-confidence Guarantees* (FRG) framework in Section 4. After proving that FRG ensures fairness for all downstream models and tasks with high probability in Section 5, we present empirical evaluations that demonstrate FRG's effectiveness at upper bounding unfairness for multiple downstream models and tasks in Section 7.

## 2 BACKGROUND

In this section, we introduce the notation used for representation learning, define a measure of unfairness for classification models, and review a useful property relevant to fair representation learning.

### 2.1 NOTATION FOR REPRESENTATION LEARNING

Let $X$ be a random variable called the *feature vector*, $S$ be a random variable called the *sensitive attribute*, and $D := \{(X_i, S_i)\}_{i=1}^n$ be the *dataset*, where $X_1 \ldots X_n$ are i.i.d. random variables with the same distribution as $X$, $S_1 \ldots S_n$ are i.i.d. random variables with the same distribution as $S$, and each $(X_i, S_i)$ has the same joint distribution as $(X, S)$. We define $\mathcal{D}$ to be the set of all possible $D$. Let $\phi \in \Phi$ be the *representation model parameters* and $q_\phi$ be the *representation model* parameterized by $\phi$. Then, we define $Z$ to be the *representation* for $(X, S)$ where $Z \sim q_\phi(\cdot|X, S)$ and $Z \in \mathbb{R}^l$.

We assume that the learned representation will be used for subsequent supervised learning tasks, which we call *downstream tasks*. We denote the *label* for such a downstream task as the random variable $Y$. The objective in a downstream task is to predict $Y$ given $(X, S)$. Instead of using $(X, S)$ directly as input, we use $Z$ as input to a *downstream model* $\tau : \mathbb{R}^l \to \mathbb{R}$. Let $\hat{Y} := \tau(Z)$ denote the prediction of $Y$ produced by model $\tau$. We call $\hat{Y}$ the *downstream prediction*. Notice that different downstream tasks correspond to different joint distributions of $(X, S, Y)$, but we assume all downstream tasks share the same joint distribution of $(X, S)$. Thus, the same representation $Z$ can be used for multiple downstream tasks.

### 2.2 A MEASURE OF UNFAIRNESS FOR DOWNSTREAM MODELS

We will define a fair representation model to be one that ensures fairness for all possible downstream tasks and all possible downstream models. To achieve this, we must first establish a definition of fairness for downstream models and tasks. In this work, we focus on classification tasks and a widely used group fairness measure based on demographic parity (Dwork et al., 2012). Let $Y$ and $S$ be binary, leading to the following definition.

**Definition 2.1 (A measure of how unfair a downstream model $\tau$ is under demographic parity)** *Let $\Delta_{DP}(\tau, \phi)$ denote a measure of how unfair downstream predictions $\hat{Y}$ are when using representation parameters $\phi$ and downstream model $\tau$. Specifically,*

$$\Delta_{DP}(\tau, \phi) := \Big| \Pr(\hat{Y} = 1|S = 1) - \Pr(\hat{Y} = 1|S = 0) \Big|. \tag{1}$$

When $S$ is non-binary, $\Delta_{\text{DP}}(\tau, \phi)$ is defined as the maximum absolute difference between the conditional probabilities, $\Pr(\hat{Y} = 1|S)$, with any pair of values of $S$ (Bird et al., 2020).

## 2.3 MUTUAL INFORMATION BOUNDS DEMOGRAPHIC PARITY

The demographic-parity-based measure is specified for downstream models. However, we want our representation model to guarantee fairness for every possible downstream model and downstream task. Gupta et al. (2021) derived a bound for $\Delta_{\mathrm{DP}}(\tau, \phi)$ that removes the dependency on the downstream model $\tau$. Specifically, Gupta et al. (2021) showed that the mutual information between the representation and the sensitive attributes, denoted by $I(Z; S)$, can be used to limit the demographic parity of downstream models.

**Property 2.2 (Relation of mutual information to $\Delta_{\mathbf{DP}}(\tau, \phi)$)** *For all downstream models $\tau$ in all downstream tasks,*

$$I(Z; S) \geq \psi(\Delta_{DP}(\tau, \phi)),$$

*where $\psi$ is a strictly increasing non-negative convex function derived by Gupta et al. (2021), and the details of which are in Appendix A.2.* **Proof.** *See the work of Gupta et al. (2021).*

Notice that Property 2.2 does not provide a direct upper bound on $\Delta_{\mathrm{DP}}(\tau, \phi)$. Instead, it provides an upper bound on a strictly increasing non-negative convex function of $\Delta_{\mathrm{DP}}(\tau, \phi)$. We use this property later to guarantee fairness for representation models with high confidence (see Sec. 4).

## 3 PROBLEM STATEMENT

In this section, we define what it means for a representation model to be fair. We then formulate the problem of learning representation models with high-confidence fairness guarantees.

### 3.1 THE DEFINITION OF FAIR REPRESENTATION MODELS

A fair representation model should ensure with high confidence that the representations it generates will not lead to unfairness for downstream tasks. Specifically, a representation model is fair if and only if it results in fair predictions (as defined in Def. 1) for every possible downstream model and downstream task. That is, for all downstream tasks and all $\tau$, $\Delta_{\mathrm{DP}}(\tau, \phi)$ must be upper-bounded by a small constant, $\epsilon$. Formally, we define an "$\epsilon$-fair" representation model as follows.

**Definition 3.1 ("$\epsilon$-fair" representation model)** *Representation model $q_\phi$ is "$\epsilon$-fair" with parameter $\epsilon \in [0, 1]$ if and only if $\Delta_{DP}(\tau, \phi) \leq \epsilon$, for every downstream model $\tau$ and downstream task.*

### 3.2 PROBLEM FORMULATION

We define a representation learning algorithm $a : \mathcal{D} \to \Phi$ to be an algorithm that takes as input a data set and produces as output representation model parameters. In this paper, we aim to provide a representation learning algorithm such that any representation model it learns is guaranteed to be $\epsilon$-fair under Def. 3.1, with high confidence. Such an algorithm has the following formal definition.

**Definition 3.2 (A representation learning algorithm with high-confidence fairness guarantees)** *Given $\epsilon \in [0, 1], \delta \in (0, 1)$, and a dataset $D$, a representation learning algorithm $a$ is said to provide a $1 - \delta$ confidence "$\epsilon$-fairness" guarantee if and only if*

$$\Pr\left(g_\epsilon(a(D)) \leq 0\right) \geq 1 - \delta, \tag{2}$$

*where $g_\epsilon(\phi) := \sup_\tau \Delta_{DP}(\tau, \phi) - \epsilon$.*

Observe that $q_\phi$ is an $\epsilon$-fair representation model if and only if $g_\epsilon(\phi) \leq 0$ (Def. 3.1). Therefore, any algorithm under Def. 3.2 guarantees that any representation model with parameters learned by this algorithm has at least $1 - \delta$ probability to be an $\epsilon$-fair representation model.

According to Thomas et al. (2019), such algorithms can be categorized as *Seldonian* algorithms.

**Special case: unachievable $\epsilon$-fair representation models.** In some scenarios it may not be possible for any non-degenerate algorithm to ensure fairness with the specified confidence $1 - \delta$, for example when $\epsilon$, $\delta$, and the amount of available training data are all very small. In such cases, we allow the algorithm to output *No Solution Found* (NSF) as a way of indicating that it is unable to provide the required confidence that the learned representation will be fair given the amount of data it has been provided. To indicate that it is always fair for the algorithm to return NSF, we define $g_\epsilon(\phi) = 0$. However, if an algorithm constantly returns NSF, it is of no value. We empirically evaluate the probability of returning a solution (i.e., not NSF) for our algorithm in Section 7.

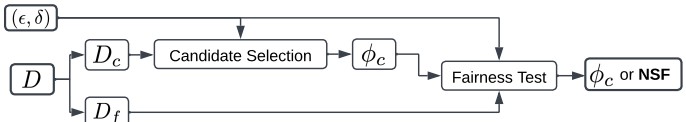

Figure 1: An overview of FRG. See Sec. 4 for discussion.

## 4 METHODOLOGY

In this section, we introduce the framework *Fair Representation learning with high-confidence Guarantees* (FRG). It is the first representation learning algorithm that provides the high-confidence fairness guarantee specified in Def. 3.2. An overview of FRG is provided in Fig. 1.

FRG consists of two major components called *candidate selection* and the *fairness test*. First we present a high-level summary of the algorithm before discussing each component in more detail. FRG first splits the data $D$ into disjoint sets, $D_c$ and $D_f$. Candidate selection uses $D_c$ to optimize and propose *candidate solution* $\phi_c$, while the fairness test uses $D_f$ to evaluate whether $\phi_c$ can satisfy $g_\epsilon(\phi_c) \leq 0$ (Def. 3.2) on future unseen data with sufficient confidence. Finally, FRG returns $\phi_c$ or NSF according to the result of the fairness test. Notice that a candidate selection algorithm that does not consider fairness may often propose candidate solutions that fail the fairness test, resulting in NSF. In the following subsections, we introduce the fairness test and then provide details for a candidate selection algorithm that proposes candidate solutions that generate representations of high expressiveness and that are likely to pass the fairness test.

### 4.1 FAIRNESS TEST

The goal of the fairness test is to evaluate whether a candidate solution $\phi_c$ induces a fair representation model with high confidence. In this section, we first develop $\tilde{g}_\epsilon(\phi)$ where $\tilde{g}_\epsilon(\phi) \leq g_\epsilon(\phi)$, and propose evaluating $\tilde{g}_\epsilon(\phi) \leq 0$ to provably determine whether a representation model $\phi$ is $\epsilon$-fair, i.e., $g_\epsilon(\phi) \leq 0$. We then propose the construction of a high-confidence upper bound on $\tilde{g}_\epsilon(\phi)$. We finally detail the evaluation process for a candidate solution $\phi_c$ using this high-confidence upper bound.

**A mutual-information-based evaluation.** Our goal is to evaluate whether $g_\epsilon(\phi) \leq 0$ with high confidence. However, estimating $g_\epsilon(\phi) = \sup_\tau \Delta_{\text{DP}}(\tau, \phi) - \epsilon$ is intractable because it requires knowledge of all downstream models and all downstream tasks. To remove the dependency on downstream models, we apply Property 2.2, and evaluate whether $I(Z; S) - \psi(\epsilon) \leq 0$ instead of $\sup_\tau \Delta_{\text{DP}}(\tau, \phi) - \epsilon \leq 0$ ($\psi$ is derived by Gupta et al. (2021) and defined in Appendix A.2). Intuitively, when the mutual information between the representation and the sensitive attribute is low, it is hard for any model to predict $S$ given $Z$ with high accuracy. Therefore, any downstream model that does not explicitly aim to predict $S$ is even less likely to take advantage of the sensitive attribute to produce unfair predictions. Theoretically, evaluating $I(Z; S) - \psi(\epsilon) \leq 0$ can provably determine the $\epsilon$-fairness of $\phi$ under Def. 3.1. We postpone the theoretical analysis to Sec. 5.

Unfortunately, computing $I(Z; S)$ is intractable because it requires marginalizing the joint distribution of $(X, S, Z)$ over feature vector $X$, and so even this approach remains intractable. Multiple previous works have derived tractable upper bounds on $I(Z; S)$, which we discuss in detail in Appendix D. Let $\tilde{I}(Z; S)$ be one of these tractable upper bounds on $I(Z; S)$. Then, we define

$$\tilde{g}_\epsilon(\phi) := \tilde{I}(Z; S) - \psi(\epsilon). \tag{3}$$

With this upper bound, we now evaluate the $\epsilon$-fairness of $\phi$ by evaluating $\tilde{g}_\epsilon(\phi) \leq 0$. In Lemma 5.1, we prove if $\Pr(\tilde{g}_\epsilon(a(D)) \leq 0) \geq 1 - \delta$, then algorithm $a$ provides the desired high-confidence fairness guarantee.

$1 - \delta$ **confidence upper bound on** $\tilde{g}_\epsilon(\phi)$**.** We follow two steps to compute a $1 - \delta$ confidence upper bound on $\tilde{g}_\epsilon(\phi)$ (if this high-confidence upper bound is at most zero, then we can conclude that $\tilde{g}_\epsilon(\phi) \leq 0$ with confidence $1 - \delta$). (1) Obtain $m$ i.i.d. unbiased estimates $\hat{g}^{(1)}, \ldots, \hat{g}^{(m)}$ of $\tilde{g}_\epsilon(\phi)$ using $D_f$, i.e., $\mathbb{E}[\hat{g}^{(j)}] = \tilde{g}_\epsilon(\phi)$ for any $j \in [1, ..., m]$. (2) Apply standard statistical tools such as Student's t-test (Student, 1908) or Hoeffding's inequality (Hoeffding, 1963) to construct a $1 - \delta$ confidence upper bound on $\tilde{g}_\epsilon(\phi)$ using $\hat{g}^{(1)}, \ldots, \hat{g}^{(m)}$. Note that we use Student's t-test for our experiments (Sec. 7).

We define $U_\epsilon : (\Phi, \mathcal{D}) \to \mathbb{R}$ to be such a function that produces a $1 - \delta$ confidence upper bound. Specifically, for $U_\epsilon(\phi, D_f)$, we have the following,

$$\Pr\Big(\tilde{g}_\epsilon(\phi) \leq U_\epsilon(\phi, D_f)\Big) \geq 1 - \delta. \tag{4}$$

**Evaluation of candidate solutions.** Suppose the fairness test gets a candidate solution $\phi_c$ and $U_\epsilon(\phi_c, D_f) \leq 0$, it follows that there is at least confidence $1 - \delta$ that $\tilde{g}_\epsilon(\phi_c) \leq 0$ (Inequality 4). Then, the fairness test concludes with at least $1 - \delta$ confidence that $q_{\phi_c}$ is an $\epsilon$-fair representation model, and $\phi_c$ passes the test. If, however, $U_\epsilon(\phi_c, D_f) > 0$, then the algorithm cannot conclude that $g_\epsilon(\phi_c) \leq 0$ with high confidence. Therefore, the fairness test concludes that there is not sufficient confidence that $q_{\phi_c}$ is an $\epsilon$-fair representation model, and $\phi_c$ fails the test.

Finally, if $\phi_c$ passes the fairness test, FRG outputs $\phi_c$. Otherwise, it outputs NSF. When $\phi_c$ fails the fairness test, we do not search for and test another representation model because this would result in the well known "multiple comparisons problem." In this case, each run of the fairness test can be viewed as a hypothesis test for determining whether the representation is fair with sufficient confidence.

## 4.2 CANDIDATE SELECTION

Notice that a representation learning algorithm using the fairness test mechanism as designed in Sec. 4.1 provides the desired $1 - \delta$ confidence $\epsilon$-fairness guarantee (Def. 3.2) regardless of the choice of candidate selection, as shown in Thm. 5.2. However, candidate selection is considered ineffective if most of its proposed solutions fail the fairness test. In this section, we introduce an effective design for candidate selection. This design results in candidate solutions that are both likely to pass the fairness test and optimized for high expressiveness.

### 4.2.1 PREDICTING WHETHER A CANDIDATE SOLUTION WILL PASS THE FAIRNESS TEST

The candidate selection mechanism proposes a candidate solution $\phi_c$ that it predicts will pass the fairness test. Such a prediction can be naturally made by leveraging knowledge of the exact form of the fairness test, except using dataset $D_c$ instead of $D_f$, i.e., checking whether $U_\epsilon(\phi_c, D_c) \leq 0$. However, there is one caveat. We repeatedly use the same dataset $D_c$ to construct high confidence upper bounds while searching for the candidate solution. Therefore, we may overfit to $D_c$, resulting in an overestimation of the confidence that the candidate solution will pass the fairness test. To mitigate this issue, we inflate (double the width of) the confidence interval used in candidate selection. We denote the inflated $1 - \delta$ confidence upper bound on $\tilde{g}_\epsilon(\phi_c)$ as $\hat{U}_\epsilon(\phi_c, D_c)$. Finally, we propose using the constraint $\hat{U}_\epsilon(\phi_c, D_c) \leq 0$ during candidate selection to find a candidate solution $\phi_c$ that is likely to pass the fairness test.

### 4.2.2 OPTIMIZING FOR A CANDIDATE SOLUTION WITH A CONSTRAINED OBJECTIVE

In addition to finding a candidate solution that is likely to pass the fairness test, candidate selection also favors candidate solutions that have high expressiveness, so that the representations it generates are effective for downstream tasks. We propose a candidate selection mechanism that achieves this goal without being limited to a specific learning algorithm. We support most parameterized representation learning architectures proposed in previous work, including the VAE-based methods (Kingma & Welling, 2013; Louizos et al., 2016), contrastive learning methods (Gupta et al., 2021; Oh et al., 2022), etc. In our experiments, we focus on an adaptation of VAE (Louizos et al., 2016) to construct the objective function that candidate selection optimizes. Specifically, we define $X \sim p_\theta(\cdot|Z, S)$ as the generative model for $X$ with input $(Z, S)$ parameterized by $\theta$. Let $\mathbb{KL}$ denote KL-divergence, and $p(Z)$ be a standard isotropic Gaussian prior, i.e., $p(Z) = \mathcal{N}(0, \mathbf{I})$, where $\mathbf{I}$ is the identity matrix. Overall, we define the candidate selection process as approximating a solution to the constrained optimization problem:

$$\max_{\theta, \phi} \ \mathbb{E}_{q_\phi(Z|X,S)}\Big[\log p_\theta(X|Z, S)\Big] - \mathbb{KL}\Big(q_\phi(Z|X, S)\|p(Z)\Big) \tag{5}$$

$$\text{s.t.} \ \ \hat{U}_\epsilon(\phi, D_c) \leq 0. \tag{6}$$

We propose using a gradient-based optimization to approximate an optimal solution $(\theta, \phi)$. When using gradient based optimizers, the inequality constraint can be incorporated into the objective

using the KKT conditions. That is, we find saddle-points of the following Lagrangian function:

$$\mathcal{L}(\theta, \phi; \lambda) := -\mathbb{E}_{q_\phi(Z|X,S)}\Big[\log p_\theta(X|Z,S)\Big] + \mathbb{KL}\Big(q_\phi(Z|X,S)\|p(Z)\Big) + \lambda\hat{U}_\epsilon(\phi, D_c), \quad (7)$$

where $\lambda \geq 0$ is the Lagrange multiplier.

## 5 THEORETICAL ANALYSIS

In this section we prove that FRG is a representation learning algorithm that provides the desired high confidence $\epsilon$-fairness guarantee, i.e., the probability that it produces a representation that is not $\epsilon$-fair for every downstream task and model is at most $\delta$.

We prove this claim in two steps. We first prove in Lemma 5.1 that if an algorithm $a$ satisfies $\Pr\left(\tilde{g}_\epsilon(a(D)) \leq 0\right) \geq 1 - \delta$, then algorithm $a$ provides the $1 - \delta$ confidence $\epsilon$-fairness guarantee described in Def. 3.2. We then prove in Theorem 5.2 that FRG indeed satisfies $\Pr\left(\tilde{g}_\epsilon(a(D)) \leq 0\right) \geq 1 - \delta$. Altogether, we can conclude that FRG guarantees with $1 - \delta$ confidence that $\Delta_{\text{DP}}(\tau, a(D))$ is upper-bounded by $\epsilon$ for any $\tau$ (recall that here $a$ corresponds to FRG and $a(D)$ corresponds to the representation model parameters returned by FRG when run on dataset $D$).

**Lemma 5.1** *If algorithm $a$ satisfies $\Pr\left(\tilde{g}_\epsilon(a(D)) \leq 0\right) \geq 1 - \delta$, then algorithm $a$ provides the $1 - \delta$ confidence $\epsilon$-fairness guarantee described in Def. 3.2.* **Proof.** *See Appendix B.*

**Theorem 5.2** *FRG provides a $1 - \delta$ confidence $\epsilon$-fairness guarantee.* **Proof.** *See Appendix C.*

## 6 PRACTICAL CONSIDERATIONS FOR UPPER BOUNDING $\Delta_{\text{DP}}$

So far we have discussed using mutual information to upper bound $\Delta_{\text{DP}}$ (the violation of the demographic parity constraint), and ensure the $\epsilon$-fairness of a representation model with high confidence (Sec. 4.1). Since $I(Z;S)$ is intractable, in Appendix D we review four tractable upper bounds on $I(Z;S)$, and also discuss why in our experiments we adopt the first upper bound, $\tilde{I}_1(Z;S)$, derived by Song et al. (2019, Section 2.2). We then test whether $\tilde{I}_1(Z;S) \leq \psi(\epsilon)$ to obtain the desired fairness guarantee (Eq. 3).

Because mutual information can be intractable, one might consider alternative methods for bounding $\Delta_{\text{DP}}$. In Appendix E, we explore potential alternatives for upper bounding $\Delta_{\text{DP}}$ but find limitations that prevent the adoption of these methods in FRG. Hence, we return to the original idea of using mutual information, $I(Z;S)$, to limit $\Delta_{\text{DP}}$. However, using mutual information has another drawback we must overcome. There tend to be significant gaps between $I(Z;S)$ and $\psi(\sup_\tau \Delta_{\text{DP}}(\tau, \phi))$, and between $\tilde{I}_1(Z;S)$ and $I(Z;S)$ (demonstrated with an example in Appendix Fig. 4). Hence, using the $\psi$-based bound on $\tilde{I}_1(Z;S)$ results in exceedingly conservative bounds on $\Delta_{\text{DP}}$. We analyze the gap between $I(Z;S)$ and $\psi(\sup_\tau \Delta_{\text{DP}}(\tau, \phi))$ in Appendix F, and the gap between $\tilde{I}_1(Z;S)$ and $I(Z;S)$ in Appendix G. Based on these analyses, one might consider using a constraint of the form

$$I(Z;S) \leq \psi(\epsilon) + \gamma + \upsilon, \quad (8)$$

where $\gamma \geq 0$ and $\upsilon \geq 0$. If $\gamma$ lower bounds the gap between $I(Z;S)$ and $\phi(\epsilon)$, and $\upsilon$ lower bounds the gap between $\tilde{I}_1(Z;S)$ and $I(Z;S)$, then $\tilde{I}_1(Z;S)$ upper bounds $\psi(\Delta_{\text{DP}}(\tau, \phi))$. One can then let

$$\tilde{g}'_\epsilon(\phi) := \tilde{I}_1(Z;S) - (\psi(\epsilon) + \gamma + \upsilon), \quad (9)$$

and use FRG to ensure $\tilde{g}'_\epsilon(\phi) \leq 0$, i.e., $\tilde{I}_1(Z;S) \leq \psi(\epsilon) + \gamma + \upsilon$, with high probability.

However, we do not know which values of $\gamma$ and $\upsilon$ satisfy $\gamma \leq I(Z;S) - \psi(\epsilon)$ and $\upsilon \leq \tilde{I}_1(Z;S) - I(Z;S)$. While one might consider treating $\gamma, \upsilon$ as tunable hyperparameters, doing so would compromise the algorithm's ability to provide a high-confidence guarantee that $\sup_\tau \Delta_{\text{DP}}(\tau, \phi) \leq \epsilon$. In summary, excluding $\gamma, \upsilon$ can make it difficult and sometimes impossible for any algorithm to return $\epsilon$-fair solutions with the desired confidence. However, including $\gamma, \upsilon$ (as hyperparameters) in our method prevents it from providing the high-confidence fairness guarantee as defined in Def. 3.2.

We provide one way to select values for $\gamma$ and $\upsilon$ that likely lower bound the mutual information gaps, as detailed in Appendix. H. Additionally, in Sec. 7.2, we present empirical evaluations to demonstrate that FRG with these practical adjustments tends to satisfy $\Delta_{\text{DP}}(\tau, \phi) \leq \epsilon$, although the formal guarantee is no longer ensured.

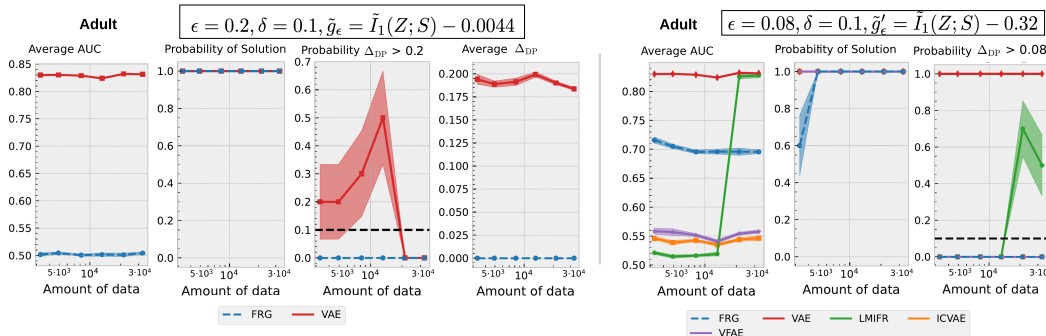

Figure 2: On the *left*, we give an example of employing FRG to provide high-confidence fairness guarantees (Def. 3.2) on the *Adult* dataset, including VAE as a baseline. On the *right*, we show the comparisons between baselines and FRG with the practical adjustments (Sec. 6) on *Adult*. The first three plots on both sides are: (1) the average AUC when applying the representations to the designated downstream task, (2) the fraction of trials that returned a solution excluding NSF, and (3) the fraction of trials that violates $\Delta_{\mathrm{DP}}(\tau, \phi) \leq \epsilon$ on the ground truth dataset. The fourth plot on the left shows the average $\Delta_{\mathrm{DP}}$ on the downstream task.

## 7 EXPERIMENTS

**Datasets.** We use 2 real-world datasets: UCI *Adult*'s sensitive attribute is *gender* and a downstream task predicts income; *UTK-Face*'s sensitive attribute is *ethnicity* and downstream labels are gender and age. We include detailed descriptions in Appendix I and the statistics in Appendix Table 1.

**Evaluations.** We are interested in addressing three research questions when evaluating FRG. (1) Do the empirical results align with our expectation that FRG produces $\epsilon$-fair representation models? In other words, is $\Delta_{\mathrm{DP}}$ of all downstream models and tasks upper-bounded by a desired $\epsilon$ with high probability? To address this question, we estimate the probabilities of violating the constraint $\Delta_{\mathrm{DP}}(\tau, \phi) \leq \epsilon$ using results from multiple runs of the algorithm with different training samples. (2) Can FRG learn expressive representations that are useful for downstream predictions? We evaluate the prediction performance on downstream tasks using the area under the ROC curve (AUC). (3) Would FRG frequently result in NSF to avoid unfairness even when a sufficient amount of data and reasonable values of $\epsilon$ and $\delta$ are provided? To address this question, we evaluate the probability that FRG provides a solution other than NSF.

**Experiment setup.** For each dataset, we hold out 20% of the data as the *test dataset*. To assess the probabilities of $\Delta_{\mathrm{DP}}$ being bounded by $\epsilon$ for all downstream models, we generate multiple training datasets from the remaining 80% of the data and train multiple representation models on these datasets. To construct the training datasets, we initially generate 10 datasets by resampling 80% of the data (excluding the test dataset) with replacement. Next, we extract proportions of 10%, 15%, 25%, 40%, 65% or 100% from each resampled dataset above to create a single *training dataset*. In total, we apply each algorithm to 60 training datasets, resulting in 60 representation models. Subsequently, we assess the results using all these models on the test dataset.

For the remainder of this section, we start by providing an example of using FRG to find representation models with high-confidence fairness guarantees. We then apply the practical adjustments (as described in Sec. 6) to FRG, and compare with competitive baselines. We conduct ablation studies on FRG with these adjustments in Appendix L.[1]

### 7.1 EVALUATION ON FRG THAT PROVIDES HIGH CONFIDENCE FAIRNESS GUARANTEE

We evaluate FRG that provides high-confidence fairness guarantees (Def. 3.2) on the *Adult* dataset. For demonstration purposes, we select $\epsilon = 0.2$ and $\delta = 0.1$, which means that FRG guarantees with 90% confidence that downstream models do not violate $\Delta_{\mathrm{DP}}(\tau, \phi) \leq 0.2$. It is worth noting that the selected $\epsilon = 0.2$ is smaller than both the $\Delta_{\mathrm{DP}}$ calculated with the true labels (as shown in Appendix Table 1), and the upper bound on $\Delta_{\mathrm{DP}}$ calculated with the prediction labels from a predictor that achieves equalized odds (Zhao et al., 2020, Theorem 3.1). We estimate $\Pr(S = 1) \approx 0.668$ from the dataset, which yields $\psi(\epsilon) \approx 0.0044$. We incorporate the constraint $\tilde{I}_1(Z; S) \leq \psi(\epsilon)$ to guarantee $\epsilon$-fairness with $1-\delta$ confidence ($\tilde{I}_1(Z; S)$ denotes the upper bound to $I(Z; S)$ as derived by Song et al. (2019, Section 2.2)). We include a vanilla VAE without any fairness consideration as a baseline.

---

[1]For Figures 3–4, each panel shows the mean (point) and standard deviation (shaded region) of a quantity as a function of the amount of data (in log scale). The black dashed line is set at the desired confidence level $\delta$.

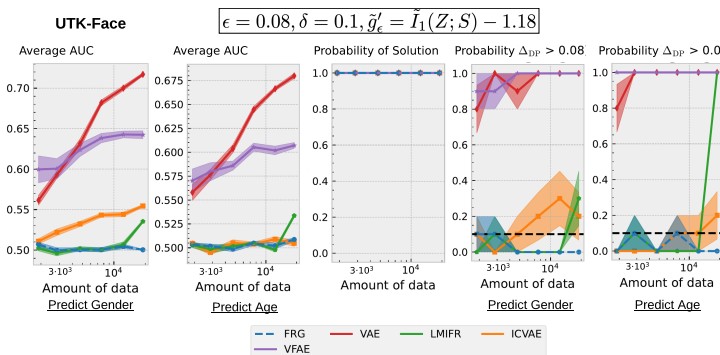

Figure 3: We show the comparisons between baselines and FRG with the practical adjustments (Sec. 6) on *UTK-Face*. Note that there are two downstream tasks, predicting gender and age. Since a learned representation model is used by all downstream tasks, there is only one probability of returning a solution excluding NSF.

We show the result in Fig. 2 Left. As demonstrated in the third and fourth plots, FRG violates the constraint $\Delta_{\mathrm{DP}}(\tau, \phi) \leq 0.2$ with a probability smaller than $0.1$, whereas VAE violates the constraint with a probability larger than $0.1$ when it uses less than $65\%$ of the training data. According to the second plot, FRG can also return solutions (i.e., not NSF) for all the trials.

Nonetheless, as discussed in Sec. 6, the constraint $\tilde{I}_1(Z; S) \leq \psi(\epsilon)$ is overly conservative, which leads to relatively low AUC on average, as illustrated in the first plot. Additionally, the fourth plot demonstrates FRG's ability to consistently keep $\Delta_{\mathrm{DP}}$ near zero. Hence, applying an even stricter $\epsilon$ constraint on FRG for high-confidence fairness guarantees is impractical and unnecessary.

## 7.2 EVALUATION ON THE PRACTICAL ADJUSTMENTS ON FRG

Considering the impracticality of strictly adhering to Def. 3.2 for FRG to provide high-confidence fairness guarantees, we evaluate FRG with the practical adjustments (as detailed in Sec. 6) that guarantees the constraint $\tilde{I}_1(Z; S) \leq \psi(\epsilon) + \gamma + \upsilon$ is satisfied with high confidence. We compare it with several baselines including LMIFR (Song et al., 2019), ICVAE (Moyer et al., 2018), VFAE (Louizos et al., 2016) and vanilla VAE. Detailed descriptions of these baselines are provided in Appendix J.

We set $\epsilon = 0.08, \delta = 0.1$ for evaluations on both the Adult and the UTK-Face datasets. For FRG, We estimate $\phi(\epsilon) + \gamma + \upsilon$ according to Appendix H and arrive at $\tilde{g}'_{\epsilon}(\phi) = \tilde{I}_1(Z; S) - 0.32$ for *Adult*, and $\tilde{g}'_{\epsilon}(\phi) = \tilde{I}_1(Z; S) - 1.18$ for *UTK-Face*. We note that as FRG does not restrict the choices of architectures for maximizing representation models' expressiveness and downstream prediction performance, we ensure fair comparisons by adopting the VAE-based primary objective as proposed in Sec. 4.2.2, and a multilayer perceptron (MLP) with one hidden layer as the downstream model across all baselines. We find hyperparameters that achieve $\Pr(\Delta_{\mathrm{DP}}(\tau, \phi) > \epsilon) < \delta$ while maximizing AUC on one designated downstream task. Detailed procedures for hyperparameter tuning are provided in Appendix K. We show the results for Adult in Fig. 2 Right, and UTK-Face in Fig. 3.

For both datasets, FRG can limit the probabilities that $\Delta_{\mathrm{DP}}(\tau, \phi) > 0.08$ to be at most $0.1$ while maintaining AUC values comparable to those of baselines. FRG also exhibits a high probability of returning solutions, excluding NSF, in most cases. The only exception occurs with the Adult dataset when data size is only $10\%$ of the total training data. Our hypothesis is that with a small amount of training data, it becomes challenging for candidate selection to recommend candidate solutions that satisfies $\tilde{g}'_{\epsilon}(\phi) \leq 0$ with high probability while simultaneously achieving high AUC.

On the Adult dataset, among the methods that upper bound the probabilities of violating $\Delta_{\mathrm{DP}}(\tau, \phi) \leq 0.08$ by $1 - \delta$, FRG achieves the highest AUC. Although LMIFR (evaluated with $65\%$ and $100\%$ of the training data) and VAE achieve higher AUC, they are more likely to violate the $\epsilon$-fairness constraint. In all other baseline evaluations, the need to tune hyperparameters to ensure high probabilities of $\Delta_{\mathrm{DP}}(\tau, \phi) \leq 0.08$ often results in subpar predictions in terms of AUC.

On UTK-Face, achieving $\epsilon$-fairness is more challenging with a multinomial and imbalanced sensitive attribute, especially when $\epsilon = 0.08$. Note that we tune hyperparameters and estimate $\tilde{g}'_{\epsilon}(\phi)$ (Appendix H) using only the gender labels while keeping the age labels hidden. The result shows that FRG can indeed effectively upper bound $\Delta_{\mathrm{DP}}$ by a small $\epsilon$ across multiple downstream tasks

with a high probability. Although baseline methods can occasionally achieve higher AUC, and IC-VAE and LMIFR can maintain $\Delta_{\mathrm{DP}} \leq 0.08$ most of the time, they cannot consistently ensure $\Delta_{\mathrm{DP}}$ remains below 0.08 on either downstream task with sufficient probabilities. This highlights the importance of providing high-confidence fairness guarantees when training fair representation models.

## 8 RELATED WORK

Fair representation learning (FRL) has been studied since at least 2013 (Zemel et al., 2013). While some prior works (Lahoti et al., 2019b;a; Peychev et al., 2022; Ruoss et al., 2020) focus on individual fairness, we focus on group fairness with unfairness quantified by metrics like demographic parity, equalized odds, equal opportunity, and others (Dwork et al., 2012; Hardt et al., 2016).

Several FRL studies prioritize optimization with respect to a specific downstream task (Calmon et al., 2017; McNamara et al., 2017; Zehlike et al., 2019; Calmon et al., 2018; Gordaliza et al., 2019; Shui et al., 2022; Zhu et al., 2021; Rateike et al., 2022). In this paper, we instead focus on learning general representations that are fair, even when downstream tasks are unknown or unlabeled.

Numerous related studies aim to achieve FRL without relying on labels from downstream tasks (Hort et al., 2022; Mehrabi et al., 2021). One category of these methods draws inspiration from information theory and probability theory, focusing on either reducing the mutual information between sensitive attributes and representations (Song et al., 2019; Jaiswal et al., 2020; Kairouz et al., 2022; Kim & Cho, 2020; Liu et al., 2022b; Moyer et al., 2018; Gupta et al., 2021), or maximizing the conditional entropy of sensitive attributes given representations (Xie et al., 2017; Roy & Boddeti, 2019; Sarhan et al., 2020). One work explores the use of distance covariance as an alternative to mutual information (Liu et al., 2022a). Some methods can limit downstream unfairness by constraining the total variation distance between the representation distributions of different groups (Madras et al., 2018; Zhao et al., 2020; Shen et al., 2021; Balunović et al., 2022). Other approaches promote independence among sensitive attributes through penalization based on Maximum Mean Discrepancy (Louizos et al., 2016; Oneto et al., 2020a; Deka & Sutherland, 2023), adversarial training that limits the adversary's performance in predicting sensitive attributes (Edwards & Storkey, 2016; Feng et al., 2019; Qi et al., 2022; Wu et al., 2022; Kim et al., 2022), meta-learning (Oneto et al., 2020b), PCA (Lee et al., 2022; Kleindessner et al., 2023), measures for statistical dependence (Grari et al., 2021; Quadrianto & Sharmanska, 2019), learning a shared feature space between groups (Cerrato et al., 2021), or disentanglement (Creager et al., 2019; Oh et al., 2022; Locatello et al., 2019).

Some FRLs provide theoretical analyses. Madras et al. (2018); Gupta et al. (2021); Zhao et al. (2020); Shen et al. (2021) offer provable upper bounds on the unfairness of all downstream models and tasks. Gitiaux & Rangwala (2021) present a method to compute an empirical upper bound on the expected values of unfairness for all downstream models and tasks. Some recent work (Jovanović et al., 2023; Balunović et al., 2022) provides practical certificates that serve as high-confidence upper bounds on the unfairness, using finite samples, but their methods are limited to particular representation model architectures (e.g., decision trees, normalizing flows) and representation distributions (e.g., discrete), which may not be ideal in some situations. Furthermore, these methods do not provide a framework for learning fair representation models with high confidence. Specifically, they do not accept a user-defined threshold and ensure with high confidence that the unfairness of all downstream models is upper-bounded by that threshold.

Finally, some prior work (Li et al., 2022; Thomas et al., 2019; Hoag et al., 2023) provides high-confidence guarantees for fair classification, but does not explore the representation learning setting.

## 9 CONCLUSION AND FUTURE WORK

In this work, we introduced FRG, a fair representation learning framework that provides high-confidence fairness guarantees, ensuring that unfairness for all downstream models and tasks is upper-bounded by a user-defined threshold. After substantiating our work with theoretical analysis, we conducted empirical evaluations that demonstrate FRG's effectiveness in upper-bounding unfairness across various downstream models and tasks. In the future, we plan to extend FRG for representation learning with other guarantees related to privacy, safety, robustness, and more.

## 10 REPRODUCIBILITY STATEMENT

We provide the source code in an anonymous repository here[2] and in the `zip` file in Supplementary Material. The detailed instructions for creating the environment, acquiring the datasets, and running FRG and baselines are presented in `README.md`. One may refer to Section 7 to get the details of experiment setup and may refer to Appendix I, J, K to get the details of the datasets, the baselines and the hyperparameter tuning procedure.

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

## A    DETAILS OF PROPERTY 2.2

Gupta et al. (2021) has derived Property 2.2 where $I(Z; S)$ is an upperbound for a strictly increasing non-negative convex function in $\Delta_{\text{DP}}$ of any $\tau$, which we denote as $\psi$. Gupta et al. (2021) has also found that when $I(Z; S) = 0$, $\psi(\Delta_{\text{DP}}(\tau, \phi)) = 0$ and $\Delta_{\text{DP}}(\tau, \phi) = 0$. We now define $\psi$ in detail by first introducing a helper function $f$.

**Definition A.1 (A helper function $f$)**

$$f(V) = \max\left(\log\left(\frac{2+V}{2-V}\right) - \frac{2V}{2+V}, \frac{V^2}{2} + \frac{V^4}{36} + \frac{V^6}{288}\right).$$

*with domain $V \in [0, 2)$.*

**Definition A.2 (function $\psi$ with parameter $\Delta_{\text{DP}}(\tau, \phi)$)** *When $S$ is binary, and $f$ follows Def. A.1,*

$$\psi(\Delta_{DP}(\tau, \phi)) = (1 - \pi)f(\pi\Delta_{DP}(\tau, \phi)) + \pi f((1 - \pi)\Delta_{DP}(\tau, \phi))$$

*where $\pi = P_s(S = 1)$ with $P_s$ as the marginal distribution of $S \in \{0, 1\}$.*

*When $S$ is multinomial with $K$ classes,*

$$\psi(\Delta_{DP}(\tau, \phi)) = f(\alpha\Delta_{DP}(\tau, \phi)), \alpha = \min_{k=1,\dots,K} \pi_k,$$

*where $\pi_k = P_s(S = k)$ with $P_s$ as the marginal distribution of $S \in \{1, \dots, K\}$.*

## B    PROOF OF LEMMA 5.1

**Lemma B.1 (Lemma 5.1 restated)** *If algorithm $a$ satisfies $\Pr(\tilde{g}_\epsilon(a(D)) \leq 0) \geq 1 - \delta$, then algorithm $a$ provides the $1 - \delta$ confidence $\epsilon$-fairness guarantee described in Def. 3.2.*

*Proof.* Suppose $\Pr(\tilde{g}_\epsilon(a(D)) \leq 0) \geq 1 - \delta$. By Eq. 3, $\tilde{g}_\epsilon(a(D)) = \tilde{I}(Z; S) - \psi(\epsilon) \geq I(Z; S) - \psi(\epsilon)$. By property 2.2, $I(Z; S) \geq \sup_\tau \psi(\Delta_{\text{DP}}(\tau, a(D)))$. So, the event $(\tilde{g}_\epsilon(a(D)) \leq 0)$ implies that$(I(Z; S) - \psi(\epsilon) \leq 0)$, which further implies $(\sup_\tau \psi(\Delta_{\text{DP}}(\tau, a(D))) - \psi(\epsilon) \leq 0)$. Using the fact that $\psi$ is strictly increasing in $[0, 1]$ (Appendix A.2), we have the following equivalent events:

$$\left(\sup_\tau \psi(\Delta_{\text{DP}}(\tau, a(D))) - \psi(\epsilon) \leq 0\right) \tag{10}$$

$$\iff \left(\psi(\sup_\tau \Delta_{\text{DP}}(\tau, a(D))) \leq \psi(\epsilon)\right) \tag{11}$$

$$\iff \left(\sup_\tau \Delta_{\text{DP}}(\tau, a(D)) \leq \epsilon\right) \tag{12}$$

$$\iff \left(\sup_\tau \Delta_{\text{DP}}(\tau, a(D)) - \epsilon \leq 0\right) \tag{13}$$

$$\iff (g_\epsilon(a(D)) \leq 0). \tag{14}$$

It follows that $\Pr(g_\epsilon(a(D)) \leq 0) \geq \Pr(\tilde{g}_\epsilon(a(D)) \leq 0) \geq 1 - \delta$. So, FRG (algorithm $a$) provides the desired $1 - \delta$ confidence $\epsilon$-fairness guarantee described in Def. 3.2, completing the proof.

## C    PROOF OF THEOREM 5.2

**Theorem C.1 (Theorem 5.2 restated)** *FRG provides the $1 - \delta$ confidence $\epsilon$-fairness guarantee described in Def. 3.2.*

*Proof.* By Lemma 5.1, if FRG satisfies $\Pr(\tilde{g}_\epsilon(a(D)) \leq 0) \geq 1 - \delta$, then FRG provides the desired $1 - \delta$ confidence $\epsilon$-fairness guarantee. We prove by contradiction that when $a$ represents FRG, $\Pr(\tilde{g}_\epsilon(a(D)) \leq 0) \geq 1 - \delta$.

We begin by assuming the result is false and then derive a contradiction. The beginning assumption is that $\Pr(\tilde{g}_\epsilon(a(D)) \leq 0) < 1 - \delta$. By contrapositive, we have $\Pr(\tilde{g}_\epsilon(a(D)) > 0) \geq \delta$. By the construction of FRG, $a(D)$ is either NSF or the proposed candidate solution $\phi_c \in \Phi$. Notice that $\tilde{g}_\epsilon(a(D)) > 0$ if and only if $a(D)$ does not return NSF but returns $\phi_c$ instead, i.e., $a(D) = \phi_c$. The fairness test in FRG returns $\phi_c$ if and only if $U_\epsilon(\phi_c, D_f) \leq 0$ (Sec. 4.1). Therefore, the following events are equivalent ($\Pr(A, B)$ denotes the joint probability of $A$ and $B$):

$$(\tilde{g}_\epsilon(a(D)) > 0) \tag{15}$$

$$\iff (\tilde{g}_\epsilon(a(D)) > 0, a(D) = \phi_c, U_\epsilon(\phi_c, D_f) \leq 0) \tag{16}$$

$$\iff (\tilde{g}_\epsilon(\phi_c) > U_\epsilon(\phi_c, D_f), a(D) = \phi_c). \tag{17}$$

The joint event $(\tilde{g}_\epsilon(\phi_c) > U_\epsilon(\phi_c, D_f), a(D) = \phi_c)$ implies $(\tilde{g}_\epsilon(\phi_c) > U_\epsilon(\phi_c, D_f))$. Therefore,

$$\Pr(\tilde{g}_\epsilon(\phi_c) > U_\epsilon(\phi_c, D_f)) \geq \Pr(\tilde{g}_\epsilon(\phi_c) > U_\epsilon(\phi_c, D_f), a(D) = \phi_c) \geq \delta.$$

However, by construction of the fairness test, $\Pr(\tilde{g}_\epsilon(\phi_c) \leq U_\epsilon(\phi_c, D_f)) \geq 1 - \delta$ (Inequality 4), which implies $\Pr(\tilde{g}_\epsilon(\phi_c) > U_\epsilon(\phi_c, D_f)) < \delta$. This gives a contradiction, completing the proof.

We note that this theorem is true for any choice of candidate selection, as the proof assumes the candidate solution $\phi_c$ is arbitrary.

# D   THE TRACTABLE UPPER BOUNDS ON $I(Z; S)$

To our best knowledge, there are four tractable upper bounds on mutual information $I(Z; S)$ as derived by previous work (Song et al., 2019; Moyer et al., 2018; Gupta et al., 2021). Next, we discuss these approaches and their limitations. Although our general approach is compatible with any upper bound on mutual information, given the limitations of each method, we consider the first of the two approaches ($\tilde{I}_1(Z; S)$ below) by Song et al. (2019) the most suitable in practice. Thus, we only adopt $\tilde{I}_1(Z; S)$ in our experiments.

Song et al. (2019) proposed two upper bounds on $I(Z; S)$.

$\tilde{I}_1(Z; S)$**: the first upper bound derived by (Song et al., 2019, Section 2.2).** We denote the first upper bound as $\tilde{I}_1(Z; S)$ and $\tilde{I}_1(Z; S) \geq I(Z; X, S) \geq I(Z; S)$ (Song et al., 2019, Section 2.2). This is a theoretically guaranteed upper bound. We discuss the limitation of this upper bound in Appendix G that using this upper bound may diminish the expressiveness of the representations. However, we still find it effective for FRG to limit $\Delta_{\mathrm{DP}}$ by $\epsilon$ in experiment (Sec. 7).

$\tilde{I}_2(Z; S)$**: the second upper bound derived by (Song et al., 2019, Section 2.3).** Song et al. (2019) proposed a tighter upper bound compared to $\tilde{I}_1(Z; S)$, which we denote as $\tilde{I}_2(Z; S)$. However, it requires adversarial training, and the true upper bound can only be obtained when the adversarial model approaches global optimality. This is not ideal because if the adversarial model is under-performing, we may under-estimate the upper bound to $I(Z; S)$, and guaranteeing $\tilde{I}_2(Z; S) \leq \psi(\epsilon)$ does not guarantee $I(Z; S) \leq \psi(\epsilon)$ or $\epsilon$-fairness. This result has also been confirmed by prior work including Xu et al. (2021); Elazar & Goldberg (2018); Gupta et al. (2021) and Gitiaux & Rangwala (2021).

$\tilde{I}_3(Z; S)$**: the upper bound derived by Moyer et al. (2018).** Moyer et al. (2018) found $I(Z; S) = I(Z; X) - H(X|S) + H(X|Z, S)$ where $H$ denotes entropy. They proposed ignoring the unknown positive constant term $H(X|S)$ and using the reconstruction error, i.e., $-\mathbb{E}_{q_\phi(Z|X,S)}\Big[\log p_\theta(X|Z, S)\Big]$ to be an upper bound of $H(X|Z, S)$ (Moyer et al., 2018, Equations 2–7). Let $\tilde{I}_3(Z; S) := I(Z; X) - \mathbb{E}_{q_\phi(Z|X,S)}\Big[\log p_\theta(X|Z, S)\Big]$. Suppose $\tilde{I}_3(Z; S) - I(Z; S) > \psi(\epsilon) + \gamma$ ($\gamma$ is defined in Inequality 8), then any $\phi$ that satisfies $\tilde{I}_3(Z; S) \leq \psi(\epsilon) + \gamma$ would result in $I(Z; S) \leq 0$, which is a constraint that is impossible to satisfy. Moreover, it can be difficult to estimate the gap $\tilde{I}_3(Z; S) - I(Z; S)$ because (1) $H(X|S)$ is hard to estimate; (2) $\tilde{I}_3(Z; S)$ is sensitive to the performance of the reconstruction model.

$\tilde{I}_4(Z; S)$**: the upper bound derived by Gupta et al. (2021).** Gupta et al. (2021) observed that $I(Z; S) = I(Z; S|X) + I(Z; X) - I(Z; X|S)$. They then derived a lower bound for the term

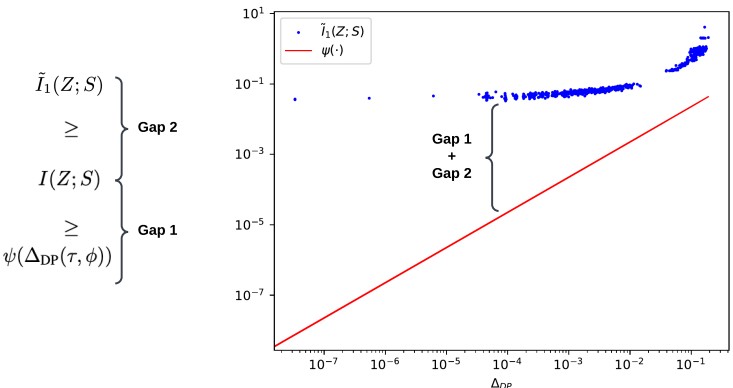

Figure 4: Using the *Adult* dataset (details in Appendix. I), we run L-MIFR (Song et al., 2019) with different hyper-parameters to find representation models that achieve different $\Delta_{\mathrm{DP}}(\tau, \phi)$. For each of the representation model, we record the corresponding tractable upper bound to $I(Z; S)$ by Song et al. (2019, Section 2.2), denoted as $\tilde{I}_1(Z; S)$, and make the scatter plot in blue. We plot the function $\psi(\cdot)$ (Appendix A.2) in red. We highlight that there exists a gap between $\tilde{I}_1(Z; S)$ and $\psi(\Delta_{\mathrm{DP}}(\tau, \phi))$, which consists of two gaps, $\tilde{I}_1(Z; S) - I(Z; S)$ and $I(Z; S) - \psi(\Delta_{\mathrm{DP}}(\tau, \phi))$, and it can be observed empirically as shown by the plot.

$I(Z; X|S)$ using constrative estimation so that $I(Z; S)$ can be upper-bounded. Specifically, they proved $I(Z; X|S) \geq \mathbb{E}_{p(X,Z,S)}\left[\log \frac{e^{f(X,Z,S)}}{\frac{1}{M}\sum_{m=1}^{M} e^{f(X_m,Z,S)}}\right]$, where $p(X, Z, S)$ is the joint distribution of $(X, Z, S)$, $X_1, \cdots, X_M \sim p_{X|S}$, $p_{X|S}$ is the conditional distribution of $X$ given $S$, and $f$ is an arbitrary function (Gupta et al., 2021, Proposition 5). Since the distribution $p_{X|S}$ is unknown, the authors use the $X, S$ pairs in the dataset as samples from this conditional distribution. When making a point estimate of the expectation, they use one sample from the dataset to evaluate the numerator, and use $M$ samples from the same dataset to evaluate the denominator. This means that the estimation of the expectation can be biased because the point estimates are not independent. Empirically, we also observe this issue and find that it tends to result in over-estimates of $I(Z; X|S)$ and under-estimates of $I(Z; S)$. Given how these terms are used in the expression for $I(Z; S)$, this results in bounds on mutual information that do not hold.

## E  ALTERNATIVE METHODS FOR UPPER-BOUNDING $\Delta_{\mathrm{DP}}$

One might consider alternative methods for bounding $\Delta_{\mathrm{DP}}$ because mutual information can be intractable and there can be a significant gap between mutual information and $\psi(\Delta_{\mathrm{DP}})$ (that is, the upper bound can be loose). Several alternative methods have been proposed, which can provide bounds on $\Delta_{\mathrm{DP}}$ using bounds on the total variation between the conditional distributions $p_{\tau,\phi}(\hat{Y}|S = 0)$ and $p_{\tau,\phi}(\hat{Y}|S = 1)$ (Zhao et al., 2020; Madras et al., 2018; Shen et al., 2021; Balunović et al., 2022). However, to our knowledge, there is not a known function such as $\psi$ (Appendix A.2) that expresses the relation between the total variation and demographic parity, so total variation cannot be used to upper bound $\sup_\tau \Delta_{\mathrm{DP}}(\tau, \phi)$ with a specific $\epsilon$. In other work, Jovanović et al. (2023, Section 5) proposed a practical certificate that upper bounds $\sup_\tau \Delta_{\mathrm{DP}}(\tau, \phi)$. However, their method requires $Z$ to be a discrete random variable, which is restrictive for general representation learning. Therefore, these methods are not suitable for our framework as they cannot be used to learn $\epsilon$-fair representation models with a high-confidence guarantee.

## F  THE NON-TRIVIAL GAP BETWEEN $I(Z; S)$ AND $\psi(\sup_\tau \Delta_{\mathrm{DP}}(\tau, \phi))$

In this section, we analyze the non-trivial gap between $I(Z; S)$ and $\psi(\sup_\tau \Delta_{\mathrm{DP}}(\tau, \phi))$ that makes it difficult for any algorithm to obtain $\epsilon$-fairness.

As shown by Gupta et al. (2021, Figure 6), there tends to be a significant gap between $I(Z; S)$ and $\psi(\sup_\tau \Delta_{\mathrm{DP}}(\tau, \phi))$. Using their Figure 6 as an example, when $I(Z; S) \approx 0.035$, $\Delta_{\mathrm{DP}}(\tau, \phi) \approx 0.15$

and $\psi(\Delta_{\text{DP}}(\tau, \phi)) \approx 0.0025$. So, to ensure that $\Delta_{\text{DP}}(\tau, \phi) \leq 0.15$ with high confidence using the $\psi$-based bound on mutual information, one must ensure that $I(Z; S) \leq 0.0025$ with high confidence. However, in reality ensuring that $\Delta_{\text{DP}}(\tau, \phi) \leq 0.15$ only requires $I(Z; S) \leq 0.035$. Obtaining a solution that satisfies $I(Z; S) \leq 0.0025$ is far more difficult than obtaining a solution that satisfies $I(Z; S) \leq 0.035$, and hence using the $\psi$-based bound on mutual information results in exceedingly conservative bounds on $\Delta_{\text{DP}}$.

# G    THE NON-TRIVIAL GAP BETWEEN $\tilde{I}_1(Z; S)$ AND $I(Z; S)$

In this section we analyze the non-trival gap between $\tilde{I}_1(Z; S)$ and $I(Z; S)$ where $\tilde{I}_1(Z; S)$ (Appendix D) is one of the upper bounds to $I(Z; S)$ as derived by Song et al. (2019, Section 2.2). We begin by analyzing the gap between $I(Z; X, S)$ and $I(Z; S)$. $I(Z; X, S) - I(Z; S) = H(Z|S) - H(Z|X, S) = H(X|S) - H(X|Z, S) = I(X; Z|S)$. This is the mutual information between $X$ and $Z$ given $S$, which is closely related to the primary objective we hope to maximize. Overall, we have the following:

$$I(Z; S) \leq I(Z; X, S) \tag{18}$$
$$= I(Z; S) + I(X; Z|S) \tag{19}$$
$$\leq \tilde{I}_1(Z; S) \tag{20}$$

Although using a constraint $\tilde{I}_1(Z; S) \leq \psi(\epsilon)$ encourages both $I(Z; S)$ and $I(X; Z|S)$ to be small which seems to diminish the expressiveness of the representation model, we show empirically that it is effective for upper bounding mutual information and the $\Delta_{\text{DP}}$ of the downstream tasks with high probability in experiment (Sec. 7).

# H    PRACTICAL ADJUSTMENTS ON FRG

In this section, we detail an approach to construct the practical constraint $\tilde{I}_1(Z; S) \leq \psi(\epsilon) + \gamma + \upsilon$ (Inequality 8), and apply the constraint on FRG so that $\Delta_{\text{DP}}$ is likely to be bounded by $\epsilon$. We note, however, that although this approach results in confidence intervals that hold with roughly the desired probability, it does not result in an actual high-confidence guarantee. Although this is not optimal, methods that tend to provide reasonable confidence intervals can often be useful even if the confidence intervals do not actually have guaranteed coverage (see, for example, common applications of Student's $t$-test to non-normal data and the use of bootstrap confidence intervals (Learned-Miller & Thomas, 2020)).

To construct the constraint $\tilde{I}_1(Z; S) \leq \psi(\epsilon) + \gamma + \upsilon$, we do not need to determine $\gamma$ and $\upsilon$ separately. We only need to determine $\gamma + \upsilon$, and we want $\gamma + \upsilon$ to under-estimate the true gap $\tilde{I}_1(Z; S) - \sup_\tau \psi(\Delta_{\text{DP}}(\tau, \phi))$ so that satisfying $\tilde{I}_1(Z; S) \leq \psi(\epsilon) + \gamma + \upsilon$ implies $\sup_\tau \psi(\Delta_{\text{DP}}(\tau, \phi)) \leq \psi(\epsilon)$, and the representation model $\phi$ is $\epsilon$-fair. We propose a practical way of estimating a value for $(\gamma + \upsilon)$ using L-MIFR as follows. (1) Utilize the candidate selection data $D_c$ to run L-MIFR various hyperparameter settings, aiming to achieve $\epsilon - c \leq \Delta_{\text{DP}}(\tau, \phi) \leq \epsilon$ on $D_c$, where $0 \leq c \leq \epsilon$ is a predetermined hyperparameter. For each value of $\Delta_{\text{DP}}(\tau, \phi)$, record the corresponding $\tilde{I}_1(Z; S)$ on $D_c$. (2) Arrange all of the $\tilde{I}_1(Z; S)$ in ascending order, and select the one associated with the representation model that achieves the best downstream prediction performance with the $k$-th percentile ($k$ is a hyperparameter). (3) Let $\gamma + \upsilon$ represent the difference between the chosen $\tilde{I}_1(Z; S)$ and $\psi(\epsilon)$. Suppose none of the tried settings of L-MIFR achieves $\Delta_{\text{DP}}(\tau, \phi) \leq \epsilon$, let FRG return NSF. We introduce the hyperparameter $c$ to prevent overly conservative estimation of $\gamma + \upsilon$ since $\epsilon - c$ serves as a lower bound for $\Delta_{\text{DP}}$. Additionally, we use the $\tilde{I}_1(Z; S)$ value from the $k$-th percentile, rather than the smallest value, to estimate $\gamma + \upsilon$ because $\tilde{I}_1(Z; S)$ also serves as an upper bound for $I(X; Z|S)$ (Appendix G). To maintain the high expressiveness of the representation models, it is essential that $I(X; Z|S)$ remains relatively large. Therefore, we estimate $\gamma + \upsilon$ using one of the smallest $\tilde{I}1(Z; S)$ values that simultaneously satisfies $\Delta DP(\tau, \phi) \leq \epsilon$ and avoids excessively reducing $I(X; Z|S)$, which could lead to poor prediction performance. Overall, we define

$$\tilde{g}'_\epsilon(\phi) = \tilde{I}_1(Z; S) - (\psi(\epsilon) + \gamma + \upsilon), \tag{21}$$

| Datasets | Sensitive (n groups) | Downstream Tasks | Data Size | Data Fractions of Each Group | $\Delta_{\text{DP}}$ of True Labels | Feature Dimensions |
|---|---|---|---|---|---|---|
| Adult | Gender (**2**) | Income | 41034 | 0.668, 0.332 | 0.260 | 117 |
| UTK-Face | Ethnicity (**5**) | Gender & Age | 23700 | 0.425, 0.191, 0.145, 0.168, 0.071 | 0.120 & 0.319 | $48 \times 48$ |

Table 1: Summary of dataset statistics. $\Delta_{\text{DP}}$ of True Labels is calculated with $|\Pr(Y = 1|S = 1) - \Pr(Y = 1|S = 0)|$ where $Y$ is the true label.

and we use FRG to ensure that $\tilde{g}'_\epsilon(\phi) \leq 0$ with high probability.

While this approach is heuristic and may not provide the high-confidence fairness guarantee defined in Def. 3.2, FRG can guarantee $\tilde{I}_1(Z; S) \leq \psi(\epsilon) + \gamma + \upsilon$ with high confidence. We show empirically in Sec. 7.2 that FRG with the practical adjustment in Sec. 6 using this estimation of $\gamma + \upsilon$ tends to satisfy $\Delta_{\text{DP}}(\tau, \phi) \leq \epsilon$.

## I    DATASETS

The dataset statistics are listed in Table 1. The first dataset is the UCI *Adult* dataset,[3] which contains information of over 40,000 adults from the 1994 US Census. The downstream task is to predict whether an individual earns more than \$50K/year with gender as the sensitive attribute.

The second dataset is *UTK-Face*,[4] which is a large-scale face dataset with over 20,000 face images with annotations of age, gender, and ethnicity. We consider ethnicity as the sensitive attribute, and we include two downstream binary prediction tasks: predicting gender and classifying age groups (above or below 30). We use the pre-processed data from Kaggle.[5]

## J    BASELINES

We include four baselines, among which three are competitive fair representation learning methods. We do not consider baselines that require supervision with a labelled downstream task, such as models proposed by Madras et al. (2018); Gupta et al. (2021), etc. We also do not include baselines that restrict the choices of the representation models and the downstream models, such as models proposed by Kim & Cho (2020); Balunović et al. (2022); Jovanović et al. (2023). We list the baselines we include for experiment with descriptions as follows:

1. *L-MIFR (Song et al., 2019)* uses Lagrangian Multipliers to encourage a representation model to satisfy constraints $\tilde{I}_1(Z; S) \leq \epsilon_1$ and $\tilde{I}_2(Z; S) \leq \epsilon_2$ where $\tilde{I}_2(Z; S)$ is an upper bound on $I(Z; S)$ that involves adversarial training(as discussed in Appendix D), $\epsilon_1$ and $\epsilon_2$ are hyperparameters.

2. *ICVAE (Moyer et al., 2018)* is a baseline that adds a regularization term $\alpha \tilde{I}_3(Z; S)$ to the primary loss, where $\alpha \geq 0$ is a hyperparameter, and $\tilde{I}_3(Z; S)$ is an upper bound on $I(Z; S)$ (as discussed in Appendix D).

3. *VFAE (Louizos et al., 2016)* is a baseline that adds an Maximum Mean Discrepancy (MMD) regularizer, which encourages statistical independence between $S$ and $Z$.

4. Finally, we include the vanilla *VAE*, which is trained solely on the proposed primary objective (Sec.4.2.2) without the constraint and does not include extra consideration for fairness.

## K    HYPERPARAMETER TUNING

In our hyperparameter tuning process, we adjust various parameters, including the step sizes of the primary objective, the Lagrange multiplier, and the adversary (for L-MIFR), the weight of the regularizers (e.g. MMD for VFAE), the number of epochs, etc. The primary objective of hyperparameter tuning is not only to find a set of hyperparameters for the algorithm that minimizes $\Delta_{\text{DP}}$.

---

[3]https://archive.ics.uci.edu/ml/datasets/Adult
[4]https://susanqq.github.io/UTKFace/
[5]https://www.kaggle.com/datasets/nipunarora8/age-gender-and-ethnicity-face-data-csv?resource=download

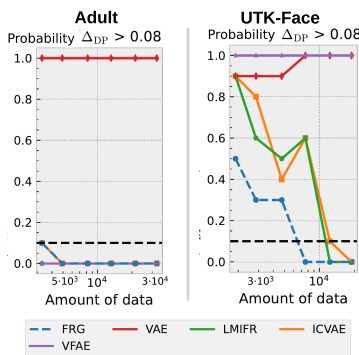

Figure 5: We show the probabilities that $\Delta_{\mathrm{DP}}(\tau, \phi) > 0.08$ evaluated on the *validation* sets (or fairness test set for FRG) for both datasets. On the Adult dataset, all baselines except VAE can maintain the probability that $\Delta_{\mathrm{DP}}(\tau, \phi) > 0.08$ to be less than $\delta = 0.1$. On the UTK-Face, when the data size is relatively large (65% and 100% of the trianing data), we find a set of hyperparameters for FRG, LMIFR, ICVAE that keeps $\mathrm{Pr}(\Delta_{\mathrm{DP}}(\tau, \phi) > 0.08) \leq \delta$. However, when the data size is small, it is difficult for all methods to keep the probability small on the validation sets.

Instead, our goal is to find hyperparameters that allow the algorithm to consistently provide a representation model that is $\epsilon$-fair with as high expressiveness as possible. Thus, one should not tune hyperparameters separately for each of the training datasets we created. When we reuse the same training or validation set for hyperparameter search, we end up evaluating $\Delta_{\mathrm{DP}}$ multiple times on the same training or validation set. As a result, $\Delta_{\mathrm{DP}}$ evaluated on the model trained with the chosen hyperparameters may provide a biased estimation of $\Delta_{\mathrm{DP}}$ on unseen future data. Consequently, the estimation of the probability $\Delta_{\mathrm{DP}} \leq \epsilon$ will also be biased. Therefore, we create additional datasets for hyperparameter tuning and adopt the same hyperparamters on different training datasets of the same size.

For baselines, we create validation sets by sampling 20% of the training data, while for FRG, we evaluate the models using the fairness test datasets (i.e., $D_f$ in Sec. 4.1). We tune each algorithm with grid search according to the metrics evaluated on the validation sets (for baselines) or on the fairness test sets (for FRG). For the UTK-Face dataset, as there are multiple downstream tasks, we only assume the gender labels are available for hyperparameter tuning. We first consider hyperparameters that yield high probabilities (at least $1 - \delta$) of satisfying $\Delta_{\mathrm{DP}} \leq \epsilon$. If the probabilities of satisfying $\Delta_{\mathrm{DP}} \leq \epsilon$ for all sets of hyperparameters are lower than $1 - \delta$, we select the sets that provide the highest probabilities while maintaining the lowest average $\Delta_{\mathrm{DP}}$. If there are ties, we prioritize the hyperparameters that achieve the highest average AUC. Note that we set the minimum allowed step size for the primary objective to $10^{-6}$. This choice is motivated by the fact that an algorithm with an excessively small step size may have minimal impact on optimizing the primary objective and could potentially produce random representations that lack utility for downstream predictions, despite being highly likely to be fair. We also note that we use the same number of dimensions for representation $Z$ and the same hidden size for the downstream MLP for fair comparison. We show the resulting $\mathrm{Pr}(\Delta_{\mathrm{DP}}(\tau, \phi) > 0.08)$ evaluated on the validation sets (or fairness test set for FRG) for both the Adult and the UTK-Face datasets in Fig. 5.

## L  ABLATION STUDY

We conduct an ablation study on FRG with the practical adjustments (Sec. 6) with two changes: (1) removing the fairness test component, (2) modifying the candidate selection process. Instead of using $\hat{U}_\epsilon(\phi, D_c) \leq 0$ as an optimization constraint, where $\hat{U}_\epsilon(\phi, D_c)$ is a predicted high-confidence upper bound on $\tilde{g}'_\epsilon(\phi)$ (Sec. 4.2), we only evaluate the expectation of $\tilde{g}'_\epsilon(\phi)$ using candidate selection data $D_c$. Formally, the ablation solves the following constrained optimization problem using the

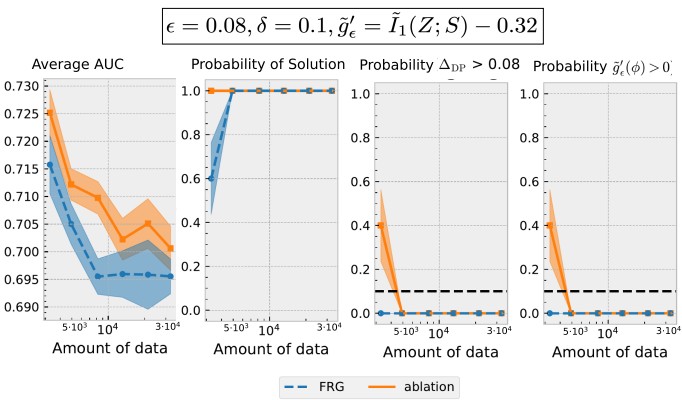

Figure 6: Ablation study on FRG with the practical adjustments (Sec. 6) using the adult dataset. We include the fourth plot which shows the probabilities the constraint $\tilde{g}'_\epsilon(\phi) \leq 0$ is violated. See Appendix L for discussion.

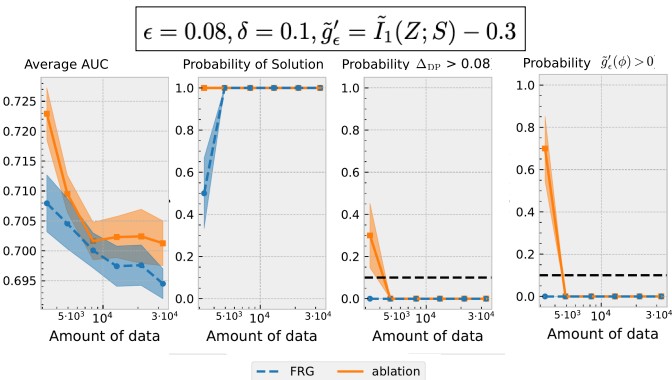

Figure 7: Ablation study on FRG with the practical adjustments (Sec. 6) using the adult dataset. Different from Fig. 6, it uses a slightly stricter constraint $\tilde{g}'_\epsilon = \tilde{I}_1(Z; S) - 0.3$. See Appendix L for discussion.

KKT conditions:

$$\max_{\theta,\phi} \quad \mathbb{E}_{q_\phi(Z|X,S)}\Big[\log p_\theta(X|Z,S)\Big] - \mathbb{KL}\Big(q_\phi(Z|X,S)\|p(Z)\Big) \tag{22}$$

$$\text{s.t.} \quad \mathbb{E}\Big[\tilde{g}'_\epsilon(\phi)\Big] \leq 0. \tag{23}$$

We conduct the experiment on the Adult dataset, and we show the results in Fig. 6.

The results indicate that, although the ablated FRG achieves a slightly better AUC, it can produce representation models that do not meet the $\epsilon$-fairness requirement due to a lack of consistent constraints on $\tilde{g}'_\epsilon(\phi) \leq 0$ with high probabilities. This highlights the necessity of a fairness test that assesses the high-confidence upper bounding of $\tilde{g}'_\epsilon(\phi)$ to ensure $\epsilon$-fairness in representation models.

We also observe that the original method has the probabilities of returning NSF (the second plot) that are equivalent to the ablated method's probabilities of violating the constraint $\tilde{g}'_\epsilon(\phi) \leq 0$ (the fourth plot). Including the fairness test for the ablated method would result in the same probability of returning NSF as the original method. This demonstrates that using the constraint $\mathbb{E}\Big[\tilde{g}'_\epsilon(\phi)\Big] \leq 0$ and the constraint $\hat{U}_\epsilon(\phi, D_c)$ for optimization in candidate selection has a similar effect in this experiment.

We use $\hat{U}_\epsilon(\phi, D_c)$ because previous work (Thomas et al., 2019; Metevier et al., 2019; Hoag et al., 2023) has demonstrated its effectiveness in preventing overfitting to training data when aiming to satisfy constraints with high probability on ground truth data. To assess whether using $\hat{U}\epsilon(\phi, D_c)$ provides an advantage, we evaluate FRG using a more stringent constraint, $\tilde{g}'\epsilon(\phi) := \tilde{I}_1(Z; S) - 0.3$, as shown in the results in Fig. 7. As expected, with the smallest data size, because of a stricter constraint, FRG is more likely to return NSF (the second plot) while the ablated FRG achieves a lower probability of $\Delta_{\text{DP}}(\tau, \phi) > 0.08$ (the third plot). However, we observe on the fourth plot that the ablated FRG violates $\tilde{g}'_\epsilon(\phi) \leq 0$ with a large probability, even larger than the probability that the original FRG returns NSF. This observation may be explained by the hypothesis that using the constraint $\mathbb{E}\Big[\tilde{g}'_\epsilon(\phi)\Big] \leq 0$ can lead to overfitting on the training data and does not provide a high probability that the constraint $\tilde{g}'_\epsilon(\phi) \leq 0$ will be satisfied on future unseen data. For future work, we will investigate whether there can be better alternative constraints than $\hat{U}_\epsilon(\phi, D_c) \leq 0$ in candidate selection to find candidate solutions that are likely to pass the fairness test while achieving better expressiveness.

