# OpenReview forum: "Learning Fair Representations with High-Confidence Guarantees"
_ICLR.cc/2024/Conference — ICLR 2024 Conference Withdrawn Submission_

### Official Review · Reviewer_YuAR · 2023-10-31

**Soundness:** 3 good
**Presentation:** 3 good
**Contribution:** 2 fair
**Rating:** 3
**Confidence:** 4

**Summary:**

This work first proposes a formal definition of fair representation with high probability. Then, the work proposes FRG, a theoretical framework for achieving the definition. The authors then provide theoretical results showing that their FRG is indeed correct. Then, after proposing several heuristics to address practical concerns of FRG, the work provides experimental verification of FRG with two datasets, showing its efficacy.

**Strengths:**

1. Surprisingly simple framework for high-confidence fair-representation learning, an important question in fairness
2. An extensive overview of recent progress in fair representation learning with sufficient motivation, even for first-time readers
3. Promising experimental results

**Weaknesses:**

1. The overall framework seems quite similar to the Seldonian algorithm design of Thomas et al. (2019), e.g., see Fig. 1 of Thomas et al. (2019)). Although it is true that Thomas et al. (2019) only considered fair classification experiments, as mentioned in this paper's related works, the proposed FRG also has an objective function related to the expressiveness of the representation, and some of the details even match; for instance, the discussions on "$1 - \delta$ confidence upper bound" on pg. 4 are quite similar to the caption of Fig. 1 of Thomas et al. (2019). Then, the question boils down to what is the novel contribution of this work, and my current understanding is that this is a simple application of Thomas et al. (2019) to fair representation learning. Of course, there are theoretical analysis, practical considerations and good experimental results that are specific to fair representation learning, but I believe that (as I will elaborate below) there are some problems that need to be addressed. Lastly, I believe that Thomas et al. (2019) should be given much more credit than the current draft.
2. Although the paper focuses on the high-probability fair representation construction (which should be backed theoretically in a solid way, IMHO), there are too many components (for "practicality") that are theoretically unjustified. There are three such main components: doubling the width of the confidence interval to "avoid" overfitting, introducing the hyperparameters $\gamma$ and $v$ for upper bounding $\Delta_{DP}$, and approximating the candidate optimization.
3. Also, the theoretical discussions can use some improvements. Although directly related to fair representation, the current theorems follow directly from the algorithm design itself and the well-known property of mutual information to $\Delta_{DP}$. For instance, I was expecting some sample complexity-type results for not returning NSF, e.g., given confidence levels, what is the sufficient amount of training data points that would not return NSF.
4. Lastly, if the authors meant for this paper to be a practical paper, then it should be clearly positioned that way. For instance, the paper should allocate much more space to the experimental verifications and do more experiments. Right now, the experiments are only done for two datasets, both of which consider binary sensitive attributes. In order to show the proposed FRG's versatility, the paper should do a more thorough experimental evaluation of various datasets of different characteristics with multiple group and/or nonbinary sensitive attributes, trade-off (Pareto front) between fairness and performance (or any of such kind), even maybe controlled synthetic experiments.


**Summary**. Although the framework is simple and has promising experiments, I believe that there is still much to be done. In its current form, the paper's contribution seems to be incremental and not clear.

**Questions:**

1. In Section 4.2.1, what does it mean for the confidence set to "overfit to $D_c$"? At least in constructing the confidence set, there is no training involved other than finding the candidate model $\phi_c$, and even then, one could use standard regularization techniques to avoid overfitting the model $\phi_c$.
2. Can one somehow incorporate bootstrapping into the framework to "avoid the overfitting," as the paper mentioned?
3. Does the framework apply to other supervised fairness, such as $\Delta_{EO}$? (Equal opportunity). For the guarantees to hold, I guess one must extend Gupta et al. (2021) somehow.

*Follow-up from Weakness 2*

4. Why double the confidence interval? Depending on the situation, couldn't one do tripling or even multiplying by some hyperparameter $c > 1$? Here, I'm concerned about what effect this brings to the overall high-probability guarantee and that this has no justification, even empirically. It would be helpful to at least provide some ablation study on this heuristic.

5. (More of a comment) The paper proposes to select $\gamma$ and $v$ using a heuristic. Although I appreciate that this allows for a strong empirical performance and the paper's Appendix H to explain in detail the design principle, the question of how this affects the claimed high-probability guarantee is not answered nor discussed in any depth. One could even go quite far to say that by introducing these heuristics, the high-probability guarantee is essentially lost, which seems to go against the central message of the paper, which is ensuring a *high-probability guarantee* on the resulting fair representation.

6. I'm concerned about how the quality of the approximated solution of the Lagrangian function (Eqn. (7)) affects the resulting high probability guarantee fairness. Let's say we have an $\varepsilon$-approximate solution. Then how would this error affect the high probability guarantee? (Of course, here, I'm not referring to the optimization landscape and its effect on the resulting quality.)

---

### Official Review · Reviewer_kxaq · 2023-11-01

**Soundness:** 3 good
**Presentation:** 2 fair
**Contribution:** 2 fair
**Rating:** 3
**Confidence:** 3

**Summary:**

The paper provides a framework for obtaining, from unlabeled data, provably "fair" representations, i.e. ones that are guaranteed to satisfy demographic parity up to error epsilon with probability delta (for user-specified epsilon and delta), for *all* downstream tasks. The authors first summarize their framework and formally give the just mentioned high-probability bounds, along with practical tweaks and improvements to make their method efficiently implementable (consisting in estimating and controlling various surrogate upper bounds on the demographic parity across all downstream tasks). They then provide experiments, in which they build VAE-based representations which indeed satisfy promised fairness guarantees on Adult and UTK-Face datasets.

**Strengths:**

1. The authors have put thought and care into describing various considerations related to instantiating their framework in practical settings (i.e., in terms of making the theoretical high-probability bound practical), in particular analyzing and drawing on solutions proposed in a significant number of existing works.
2. The provided experiments appear to demonstrate that their implementation of the proposed framework indeed obtains the claimed fairness guarantees and achieves respectable performance on the tasks considered, and does not often resort to outputting "No solution found" (which is allowed by the theoretical statement of the framework).
3. The task considered (of ensuring fairness for all possible downstream tasks) is clearly very interesting and timely.

**Weaknesses:**

Unfortunately, in its current state the paper falls short both (partially) on the experimental and, especially, on the theoretical side; this is why I cannot recommend acceptance now (and I think that too significant of a revision would be necessary to shift this opinion in this submission cycle).

(1) The paper prominently promises to conduct a theoretical analysis of their framework. Now, the framework itself is nothing but the proposal to take a bound recently proved in Gupta et al (2021), which bounds a certain function of the demographic parity error by the mutual information between the sensitive attributes and the representation, and reformulate it into a PAC-style guarantee on the demographic parity. After that, the authors note that since the bound of Gupta et al (2021) would be hard to estimate head-on (since estimating the mutual information term I(Z; S) is hard), it should be estimated in a surrogate, more conservative way, at which point they summarize several proposals for how to do that from recent literature and pick their favorite ones to be implemented in code.

Thus, given that the entire theory behind the framework has already been developed --- the main bound on the demographic parity over all downstream tasks in Gupta et al (excluding reformulating their guarantees in a PAC-style way), and the surrogate bounds on the mutual information in Song et al (2019) --- the current paper does not bring anything to the table in that regard: neither any novel surrogate bounding approaches nor any tighter theoretical guarantees that, like Gupta et al, would hold for all downstream tasks. Also, the discussion is purely restricted to demographic parity, and does not consider any other measures of unfairness, simply because that is where Gupta et al's bound applies.

(2) Given the above, the paper cannot be considered as providing a theoretical contribution to the field, so in that case it should have provided a fairly comprehensive set of experiments; if that was the case, then the abstract to the paper could have potentially stated, "We provide a set of experiments verifying the empirical usefulness of the bounds in Gupta et al in settings X, Y, Z, ..."  However, the set of experimental tasks is too far from comprehensive: it consists only of two datasets; with representations generated only through a specific natural VAE-based approach; and compares to only 4 baselines (three that promise fairness and a vanilla no-fairness VAE representation).

In particular, the authors (see Appendix J) exclude methods with clearly related goals --- such as Jovanovic et al (2023) --- on the basis, eg, that while the proposed framework gives guarantees for varied representation models and all downstream tasks, these other models include additional restrictions. Despite these additional restrictions --- often quite benign, such as, eg in the case of Jovanovic et al, requiring the representation to be discrete-valued --- the several related works listed by the authors in Appendix J are in my view quite pertinent, and should be examined, to at least give confidence that the proposed framework does not suffer losses that are too significant in exchange for "provable guarantees", and that, conversely, there are settings where these other works' proposed methods do in fact fail to provide similar fairness guarantees to the proposed method.

But also, of course, it is important to consider other representation learning models than VAEs, as well as significantly more varied/nontrivial tasks than just Adult (e.g. Folktables, and other datasets), to test the robustness of this paper's empirical conclusions.

3. The paper's writing level is overall minimally sufficient, but not quite there --- simple points like defining PAC-style guarantees and converting existing bounds into them are dwelled on for quite long in the main part, but meanwhile, important considerations such as which surrogate bounds to use, as well as experimental details, are only to be found (in scattered fashion) throughout the appendices. Moreover, some important quantities are not even formally spelled out --- eg the first surrogate bound of Song et al, which is prominently used by the authors, is just denoted by I_1 and never defined in detail in the scope of the paper.

**Questions:**

1. Please see above my comments on significantly enlarging the set of experiments as above.
2. It would appear wise to mention the crucial dependence of this paper on the bound of Gupta et al (2021) much more prominently than currently --- as of now, it is undeservedly pushed further into the paper.
3. Notation --- see above comment on I_1, and notice some other important omissions throughout the paper; such as not stating what the high-probability guarantee is over, as well as just casually stating that Hoeffding's bound or Student t test are used without spelling out either one.
4. More discussion is needed on the most related works at a more technical level than currently; e.g. zooming in on the ones listed in Appendix J. Currently, it feels like such discussions are being brushed aside rather than embraced in the paper, for reasons that these other frameworks apply in slightly more restricted or different settings.

---

### Official Review · Reviewer_pP3w · 2023-11-08

**Soundness:** 3 good
**Presentation:** 3 good
**Contribution:** 2 fair
**Rating:** 3
**Confidence:** 3

**Summary:**

The paper addresses the growing use of representation learning to generate predictive representations for various downstream tasks. The critical challenge addressed is to develop representation learning algorithms that offer strong fairness guarantees to prevent unfairness in all downstream prediction tasks. The authors rely on the results that relates mutual information with downstream unfairness by Gupta et al. (2021) to develop a framework (FRG) to generate a representation that can guarantee a bounded downstream unfairness. The authors develop a representation by training a representation algorithm and evaluating its unfairness using an upper bound of the mutual information, which is later adjusted in a data-driven manner. The results shown on two datasets (adult and UTK-face) prove FRG is able to contain demographic disparity in downstream classification models.

**Strengths:**

- The paper tackles a very important problem in the fairness literature, especially given how in many situations the data provider is not able to know how and for which task their data will be used.
- The authors do a good job at motivating the problem and summarize the existing literature.
- The paper is easy to follow and generally well written; Figure 1 is appreciated as the reader can understand the flow of FRG.
- The mathematical statements seem correct.

**Weaknesses:**

The central component of this framework relies on the result by Gupta et al. (2021), where the authors utilize existing upper bounds. This makes the evaluation of the paper necessarily focused on (a) the specifics of the framework, (b) the theoretical results, and (c) the downstream outcomes.

While I attach my questions below on both (a) and (c), my main criticism is as follows. The authors build a framework on an upper bound of an upper bound of the quantity of interest, which is further adjusted in the opposite direction with a multi-parameterized and data-dependent constant. They employ a representation checking system in which quantile estimation is arbitrarily doubled to prevent overfitting. Consequently, FRG yields downstream models with minimal predictive power (0.5 AUC, akin to random algorithms), raising concerns about its practical applicability. Given that the paper is primarily presented as a practical tool, I view these issues as potentially significant obstacles.

**Questions:**

(a) While the overall framework is clear, some components require further explanation:

- When deriving a $1-\delta$ confidence upper bound (end of page 4), how is the unbiased estimate $\hat{g}$ obtained? Are you using a bootstrap approach with $D_f$ to compute a series of $g$? Is the upper bound obtained by inverting a t-test, making a Gaussianity assumption along the way?

- My understanding is that the function $U_\epsilon$ is composed of the two steps highlighted at the end of page 4. I am concerned at this point about whether we can simply include that procedure into equation (7) and propagate the gradient through it. The authors do not seem to discuss this. Would it be easier to use a quantile regression approach for $U_\epsilon$ so that we are guaranteed to have a more tractable loss function?

- When predicting whether a candidate solution passes the fairness test, the authors correctly point out that they might be overfitting on the dataset $D_c$. However, doubling the length of the interval seems like an ad-hoc solution that merits a more in-depth discussion and justification.

(b) Section (b) makes sense to me, especially Lemma 5.1, as we are just dealing with upper bounds throughout. How does this change when we consider the practical adjustment proposed in Section 6?

(c) My main concern with FRG is that it manages to achieve no disparity because the model is not learning anything, as indicated by an AUC of 0.5, which suggests a random model. It seems like FRG returns representations that create a trivial model (always returning 0 or 1), which, by definition, has no disparity.

- For the adult dataset, FTG without the adjustment in the left series of Figures 1 consistently has an AUC of 0.5, implying that nothing is being learned from the representation. When including the adjustment, things improve, but the rightmost plot seems questionable as the threshold of $\epsilon=0.08$ appears arbitrary. I encourage the authors to use the x-axis for $\epsilon$ rather than the amount of data. In practice, modelers would typically use all available data.

- For UTK-Face, again, the method does not seem to be learning anything, with an AUC close to 0.5.

I believe the experiments should focus on the practical uses of FRG. In practice, we would want to see how the fairness-utility tradeoff changes as a function of $\epsilon$ and $\delta$. Although the data-dependent adjustment is somewhat ad-hoc, showing good tradeoffs across various settings of ($\epsilon$, $\delta$) could convince modelers to use this framework. Unfortunately, at the moment, the experiments do not seem to demonstrate that FRG produces meaningful representations for downstream learning.

---

### Author Response · Authors · 2023-11-16

We thank the reviewers for their valuable feedback. We agree our paper can be improved. We will focus on clarifying our theoretical contributions and improving empirical evaluations. Our next submission will include substantial changes based on your feedback.